# Evaluation of Satellite-Based Rainfall Estimates against Rain Gauge Observations across Agro-Climatic Zones of Nigeria, West Africa

Aminu Dalhatu Datti [1,2], Gang Zeng [1,*], Elena Tarnavsky [3], Rosalind Cornforth [3], Florian Pappenberger [4], Bello Ahmad Abdullahi [2] and Anselem Onyejuruwa [1,2]

1 Key Laboratory of Meteorological Disaster, Ministry of Education, Collaborative Innovation Center on Forecast and Evaluation of Meteorological Disasters (CIC-FEMD), Joint International Research Laboratory of Climate and Environment Change (ILCEC), Nanjing University of Information Science and Technology, Nanjing 210044, China; 202351010006@nuist.edu.cn (A.D.D.); 202251080001@nuist.edu.cn (A.O.)
2 Nigerian Meteorological Agency (NiMet), Nnamdi Azikiwe International Airport, Abuja 900105, Nigeria; a.bello@nimet.gov.ng
3 Department of Meteorology, University of Reading, Reading RG6 6UR, UK; elena.tarnavsky@reading.ac.uk (E.T.); r.j.cornforth@reading.ac.uk (R.C.)
4 European Centre for Medium-Range Weather Forecasts (ECMWF), Reading RG2 9AX, UK; florian.pappenberger@ecmwf.int
* Correspondence: zenggang@nuist.edu.cn

**Abstract:** Satellite rainfall estimates (SREs) play a crucial role in weather monitoring, forecasting and modeling, particularly in regions where ground-based observations may be limited. This study presents a comprehensive evaluation of three commonly used SREs—African Rainfall Climatology version 2 (ARC2), Climate Hazards Group Infrared Precipitation with Station data (CHIRPS) and Tropical Application of Meteorology using SATellite data and ground-based observation (TAMSAT)—with respect to their performance in detecting rainfall patterns in Nigeria at daily scales from 2002 to 2022. Observed data obtained from the Nigeria Meteorological Agency (NiMet) are used as reference data. Evaluation metrics such as correlation coefficient, root mean square error, mean error, bias, probability of detection (POD), false alarm ratio (FAR), and critical success index (CSI) are employed to assess the performance of the SREs. The results show that all the SREs exhibit low bias during the major rainfall season from May to October, and the products significantly overestimate observed rainfall during the dry period from November to March in the Sahel and Savannah Zones. Similarly, over the Guinea Zone, all the products indicate overestimation in the dry season. The underperformance of SREs in dry seasons could be attributed to the rainfall retrieval algorithms, intensity of rainfall occurrence and spatial-temporal resolution. These factors could potentially lead to the accuracy of the rainfall retrieval being reduced due to intense stratiform clouds. However, all the SREs indicated better detection capabilities and less false alarms during the wet season than in dry periods. CHIRPS and TAMSAT exhibited high POD and CSI values with the least FAR across agro-climatic zones during dry periods. Generally, CHIRPS turned out to be the best SRE and, as such, would provide a useful dataset for research and operational use in Nigeria.

**Keywords:** evaluation; satellite rainfall estimates; gauge observation; Nigeria; West Africa

## 1. Introduction

Satellite rainfall estimates (SREs) play an important role in weather monitoring, climate research, modeling and forecasting [1]. Changes in climate due to human activities have increased the frequency of hydrological extremes, such as floods and droughts, since 1980 globally, including in West Africa [2]. This has precipitated socioeconomic losses for vulnerable communities and resulted in the deaths of humans and livestock [3,4]. This

is because the communities located here are largely subsistence farmers dependent on rain-fed agriculture with only limited access to mechanization [5]. The sparseness of gauge networks has reduced the quality and precision of early warning systems, forecasting and modeling which will otherwise provide vital climate information services to the local communities [6]. A reliable and accurate alternative SRE dataset is essential to enable better weather and climate operations.

Evaluating SRE data is a prerequisite for understanding its accuracy and reliability [6]. Numerous studies have been carried out to evaluate the performance of SREs [7–10]. Aghakouchak et al. (2011) [1] evaluated CMORPH, PERSIAN, TMPA-V6, TMPA-RT and Stage IV in the Central United States (USA). They concluded that none of these products could be considered to be perfect for detecting rainfall events, and also noted that as the choice of extreme rainfall threshold increased, the products tended to worsen. Mekonnen et al. (2021) [11] evaluated different SREs over the high and low land regions of the upper wash basin, Ethiopia. They discovered that the SREs exhibited better skills in the highland areas of Ethiopia than in the lowlands. This agrees with other studies [12–16] which have also noted that the performance of SREs is highly influenced by the climatic conditions and topography of the studied region. In addition, several studies [2,17–21] evaluated the accuracy of various satellite rainfall estimates against rain gauge observation over Nigeria. These studies found that the spatial and temporal skills of these SREs vary across the agro-climatic zones. Moreover, Usman et al. (2018) [22] assessed the ability of ARC2, CHIRPS, TAMSAT, TARCAT and TRMM to reproduce rainfall trends during the period 1981–2015 over the northern part of Nigeria. Their study discovered that CHIRPS gave the best estimate in terms of providing reliable estimates of daily, decadal, monthly and seasonal rainfall amount. Over southwestern Nigeria, Akinyemi et al. (2020) [18] compared the SREs with rain gauge observations from 1998 to 2016, and the results showed a high correlation between the SREs and the rain gauge observation.

The aforementioned studies that have evaluated SREs in Nigeria mainly focus on the evaluation of the SREs' performances. However, none of the previous studies evaluated their accuracy or assessed the rainfall detection capability of these SREs in Nigeria. This study addresses this gap by also providing a guide to choosing a satellite-based rainfall estimate that best captures rainfall patterns in different agro-climatic zones of Nigeria, as operational monitoring of extreme events and forecasting of rainfall are primary concerns that support decision making related to effectively managing agricultural services in Nigeria. Directly addressing this need, the study evaluates SREs from the three most commonly used satellite-based datasets, NOAA ARC2 ($0.1° \times 0.1°$), CHIRPS ($0.05° \times 0.05°$) and TAMSAT ($0.0375° \times 0.0375°$), against ground-based rainfall observations from 2002 to 2022. These SREs are currently in use in West African countries, including Nigeria.

This study identifies candidate rainfall dataset(s) for operational monitoring and research, and assesses the performance of the SREs' detection abilities. Quantitative statistics measures such as correlation (r), root mean square error (RMSE), bias, mean error (ME), the false alarm ratio (FAR), the probability of detection (POD) and the critical success index (CSI) were used in this study [23–25]. This paper is organized into five sections. After the introduction, the second section is about the data and methods used. The third section is devoted to results. The fourth section is a discussion of the findings and suggestions for future research to improve the effectiveness of climate information services in order to support smallholder farmers in Nigeria. The fifth section is a conclusion.

## 2. Data and Methods

### 2.1. Datasets

#### 2.1.1. Satellite-Based Rainfall Estimates (SREs)

The three SREs considered here are the National Oceanic and Atmospheric Administration (NOAA) ARC2 ($0.1° \times 0.1°$), University of California, Santa Barbara (UCSB) Climate Hazards Center (CHC) CHIRPS ($0.05° \times 0.05°$) and the University of Reading's TAMSAT ($0.0375° \times 0.0375°$)) datasets.

African Rainfall Climatology version 2 (ARC2) is a daily estimate with historical records from 1983 to the present and a spatial resolution of 0.1° [26]. ARC2 has been developed to help solve challenging issues surrounding short-term temporal rainfall time series in Africa. The algorithm of the ARC2 dataset uses 3-hourly thermal infrared (TIR) temperature brightness and a 235 K threshold for the assessment of rain clouds. The TIR satellite image is used to calculate cold cloud duration (CCD) based on threshold and the conversion of CCD to rainfall is made using simple linear relationships. The dataset can be accessed at ftp.cpc.ncep.noaa.gov (accessed on 10 July 2023).

The full details of the Climate Hazards Center InfraRed Precipitation with Station data (CHIRPS) are described in [27–29]. The CHIRPS product is a blended rainfall dataset that combines satellite data from several instruments with gauge observation from WMO's GTS and other sources. The CHIRPS is quasi-global (covering latitude 50°S–50°N and longitude 180°E–180°W), at the daily, pentadal (5-days) and dekadal (10-days) scale, with historical data from 1981 to the present and a spatial resolution of 0.05°. It can be accessed at https://www.chc.ucsb.edu/data/chirps (accessed on 10 July 2023).

The Tropical Application of Meteorology using SATellite data and ground-based observation (TAMSAT) is a satellite rainfall estimator, produced operationally at the University of Reading, United Kingdom [30,31]. TAMSAT products have a spatial resolution of 0.0375° and cover the period from 1983 to the present. The TAMSAT method is based on the assumption that raining clouds are identified by the temperature of cold-cloud-top tropical storms [29] as derived from Meteosat thermal infrared images. Cold cloud duration (CCD) is then calibrated against historical gauge observation to generate seasonal and spatial calibration parameters. The TAMSAT algorithm is locally calibrated using gauge observations from most African countries. Like CHIRPS, TAMSAT uses gauges for bias adjustment and is also available at daily, pentad, and dekadal time steps. It can be accessed at https://www.tamsat.org.uk/data/ (accessed on 20 July 2023).

### 2.1.2. Rain Gauge Observations

Twenty-one years (2002–2022) of daily rain gauge observations were obtained from the archive of Nigeria's Meteorological Agency (NiMet) for 42 stations that are spatially distributed across the country (Figure 1). The records for 30, 4, 6 and 2 stations date back to the 1950, 1976, 1981 and 1997, respectively. Gauge station observations were collected in ASCII format and the stations are distributed over the region between 4.3°–13.8°N latitude and 3.42°E–14.3°E longitude. The data were carefully quality-controlled and validated by NiMet's data management unit (DMU) in line with WMO guidelines [32].

### 2.2. Methodology

The SREs evaluation in this study is based on three different agro-climatic zones, defined according to rainfall characteristics of the 42 rain gauge stations in Nigeria.

The Nigerian climate is characterized by two seasons: a long wet season with the majority of rainfall and a dry season with low or no rainfall [33]. The two seasons are influenced by the southwesterly and northeasterly winds [34]. However, rainfall in Nigeria is not uniform, the Guinea Zone receives the most rainfall followed by the Savannah Zone then the Sahel Zone [2,35]. Due to the sudden and non-linear latitudinal shift of rainfall from 5°N to 10°N, the Guinea and the Savannah Zones are characterized as bimodal rainy seasons [2], while for the Sahel Zone, this process sets the stage for the unimodal rainy season from June to October. The rainfall season goes from March to May (MAM), June to August (JJA) and September to November (SON) over the Guinea and Savannah Zones while July to September (JAS) is characterized as the peak rainfall period in the Sahel Zone. Moreover, June to September (JJAS) is the period in which all the zones experience widespread rainfall events as a result of more convective activities followed by deep monsoon flow.

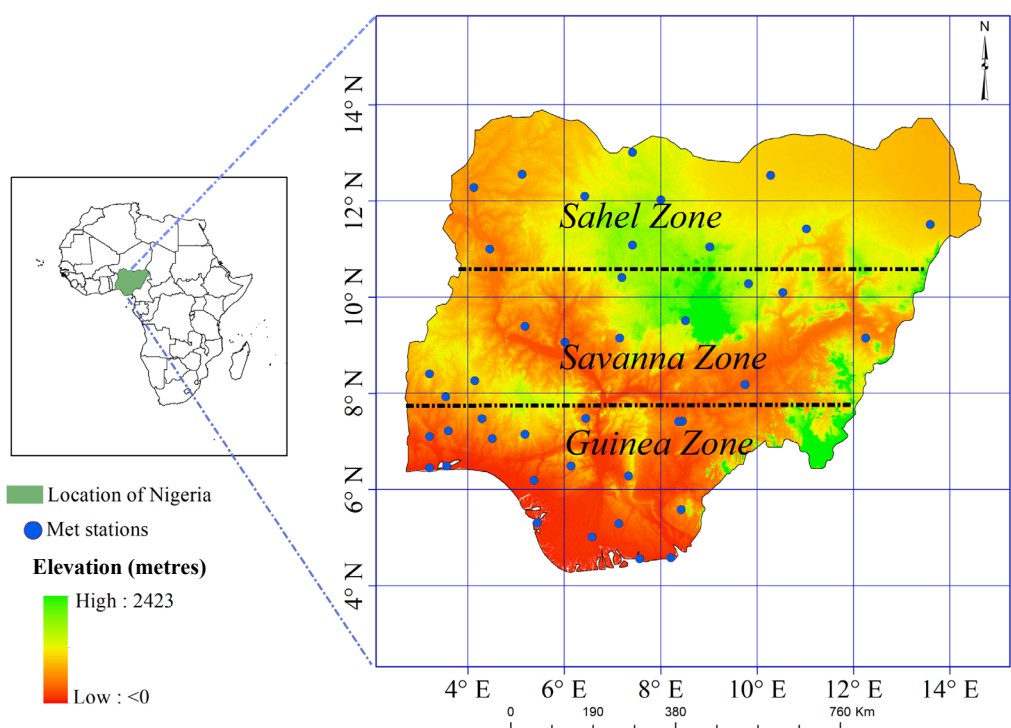

**Figure 1.** Map of Nigeria containing elevation and meteorological stations (blue dot, 11 stations are in the Sahel Zone, 12 stations in the Savannah Zone, and 19 stations over the Guinea Zone). The black dotted line represents the border line that divides each zone.

### 2.2.1. Evaluation Metrics

We extracted daily satellite rainfall pixel values for the locations corresponding to each rain gauge station (point-to-pixel) from 2002 to 2022 using the coordinates of the 42 gauge stations. This approach has been used by other studies [2,11,14,36,37] to evaluate the performance of SREs, especially in areas with poor gauge stations. A caveat is that this approach could underestimate the actual performance of satellite products [38]. Widely used evaluation metrics [29], Pearson correlation coefficient (r), root mean square error (RMSE), bias and mean error (ME) were considered in this study. The correlation coefficient (r) measures the association between two variables, and how well observed rainfall corresponds to the SREs' estimates. RMSE represents the magnitude of the estimated average error between the SREs and rain gauge observation. The smaller the value of the RMSE, the higher the central tendency. Bias shows how well SREs correspond to the rain gauge observations. A bias value that is closer to 1 indicates that the cumulative SREs are closer to the cumulative gauge rainfall observation. ME represents the average error (positive values indicate overestimation while negative values indicate underestimation). The perfect values for r, RMSE, bias and ME are $-1$ or $+1$, 0, 1 and 0 respectively [22].

$$r = \frac{\sum_{i=1}^{n}(0-\bar{O})(s-\bar{s})}{\sqrt{\sum_{i=1}^{n}(0-\bar{O})^2}\sqrt{\sum_{i=1}^{n}(s-\bar{s})^2}} \quad -1 \leq r \leq +1 \tag{1}$$

$$RMSE = \sqrt{\frac{\sum_{i=1}^{n}(S-O)^2}{n}}, \; 0 \text{ to } +\infty \tag{2}$$

$$Bias = \frac{\sum_{i=1}^{n}S}{\sum_{i=1}^{n}O} \; 0 \text{ to } +\infty \tag{3}$$

$$ME = \frac{\sum_{i=1}^{n}(S-O)}{n} \; 0 \text{ to } +\infty \tag{4}$$

where *S* and *O* are satellite and rain gauge data, s̄ and Ō are the mean satellite and rain gauge data and *n* is the total number of samples.

### 2.2.2. Categorical Skill Metrics

To understand the detection capabilities of the SREs, three categorical skill metrics such as probability of detection (POD), false alarm ratio (FAR) and critical success index (CSI) were computed [25,39,40]. These metrics are calculated based on a 2 × 2 contingency table (Table 1) [41,42]. In the contingency table, we consider rainfall rates of ≥1 mm/day [11,41,43]. The 90 quantiles (Q90) are considered to assess the capabilities of these three SREs in detecting rainfall events at higher thresholds [1,44]. POD is used to calculate the correct detected rainfall, FAR is used to calculate false events that rain gauge stations did not observe and CSI measures how accurately SREs detect rainfall events. Moreover, when the values of POD and CSI are high it simply indicates a more accurate detection ability of SREs. On the other hand, the high value of FAR indicates a high ratio of falsely detected rainfall events.

**Table 1.** Contingency table for categorical indices.

| | | Rain Gauge | |
|---|---|---|---|
| | | Yes ($R \geq x$) | No ($R < x$) |
| SREs | No ($R < x$) | Hits (H) | False alarm (FA) |
| | No ($R < x$) | Miss (M) | Correct negatives (CN) |

where *R* is the rainfall rate (mm/day) and *x* is a threshold (≥1 mm/day) for the rainfall rates [1].

$$POD = \frac{T_H}{T_{H+}T_M} \tag{5}$$

$$FAR = \frac{T_F}{T_{H+}T_{FA}} \tag{6}$$

$$CSI = \frac{T_H}{T_{H+}T_{M+}T_{FA}} \tag{7}$$

where H, M and FA are hits, misses, and false alarm detections, respectively, while $T_H$, $T_M$, and $T_{FA}$ stand for the number of times each case occurs.

## 3. Results

### 3.1. Evaluation of SREs for Estimating Rainfall

In this section, we assess the performance of all three SREs at annual and monthly time scales in order to understand their ability to estimate rainfall patterns and annual cycles by comparing their values with rain gauge data.

Figure 2 shows the spatial distribution of the mean annual rainfall estimates of ARC2, CHIRPS and TAMSAT and that observed (gauge observation) during the period 2002–2022. All three SREs showed an ability to capture spatial rainfall patterns. CHIRPS and TAMSAT show a smoother spatial rainfall pattern than ARC2. This could be due to their high spatial resolution (0.05° for CHIRPS and 0.0375° for TAMSAT). CHIRPS showed the best performance with the highest r (0.92), close to perfect bias (0.97) and lowest RMSE (61 mm/annum) (Figure 2c), followed by TAMSAT (Figure 2d) and then ARC2 (Figure 2a). It is noted that all the SREs underestimated the mean annual rainfall. Generally, the spatial rainfall patterns estimated by all the products were consistent with the observed even though CHIRPS and TAMSAT had a better resolution. This implied that all the products have the skills to capture south-north oscillations, which are partly connected to the latitudinal migration of the Intertropical Convergence Zone (ITCZ).

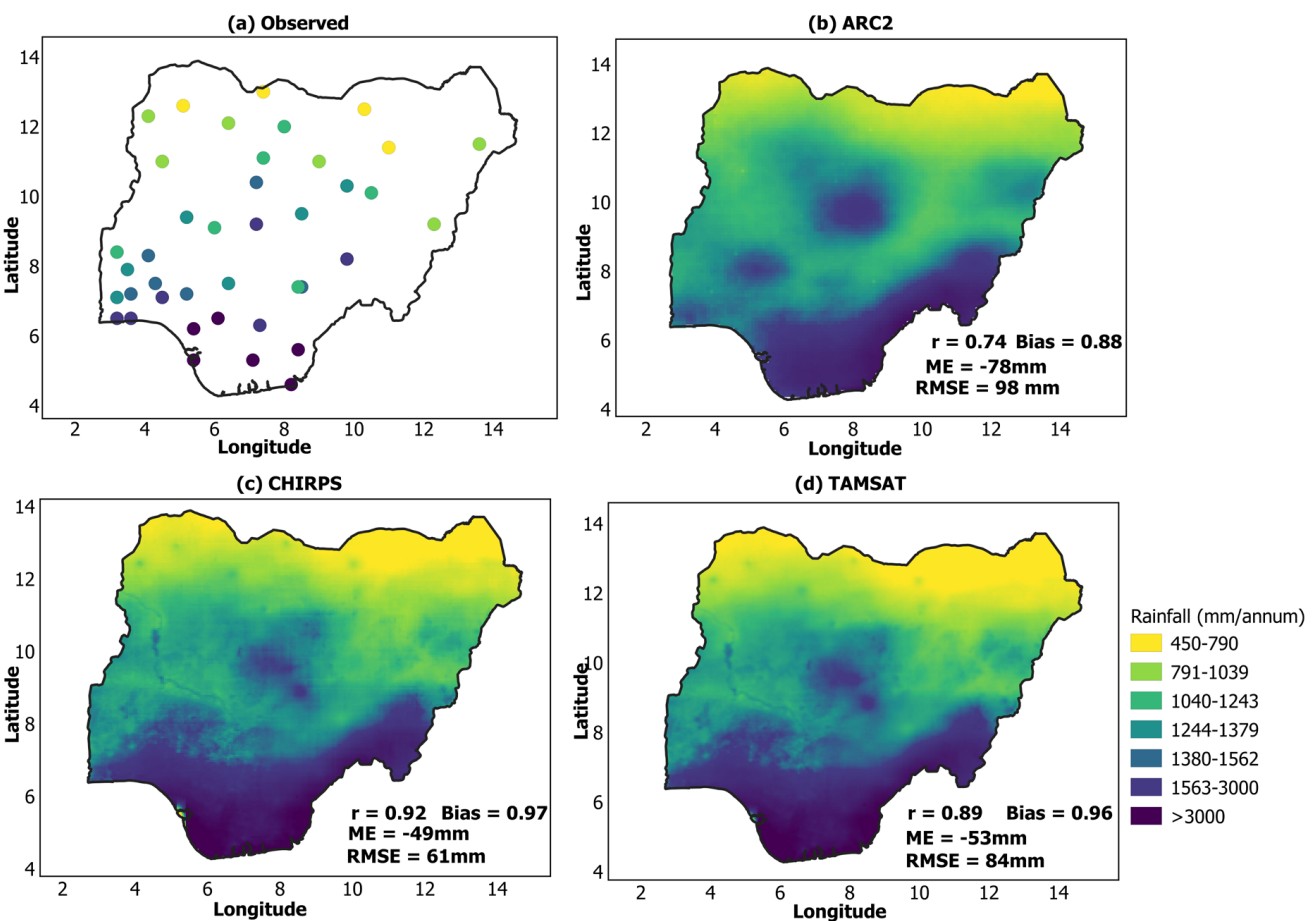

**Figure 2.** Spatial pattern of mean annual rainfall (mm/annum) of (**a**) gauge, (**b**) ARC2, (**c**) CHIRPS, and (**d**) TAMSAT for the period 2002−2022.

Figure 3a–f show the spatial and annual cycles for mean monthly rainfall over the Sahel, Savannah and Guinea Zones, respectively. A summary of the statistical indicators for the mean monthly rainfall products in all of the agro-climatic zones is presented in Table 2. The results show that all the SREs were able to capture the spatial pattern and annual cycles in all of the agro-climatic zones. This indicates that all of the satellite-derived data effectively represented the latitudinal oscillations of the ITCZ as it moved from southern to northern latitudes, bringing convective processes. Additionally, CHIRPS and TAMSAT exhibit smoother spatial rainfall patterns than ARC2 (Figure 3a–c).

Over the Sahel Zones (Figure 3d), all of the estimates exhibited a high r (>0.97). CHIRPS and TAMSAT showed an underestimation whereas ARC2 overestimated the observed rainfall. Similarly, CHIRPS and TAMSAT produced the lowest RMSE values of 4.02 mm and 10.91 mm, while ARC2 produced a high RMSE value of 16.93 mm (Table 2).

In the Savannah Zone (Figure 3e), the estimates reasonably captured the annual cycles and also indicated a strong correlation (r > 0.95), with CHIRPS producing the highest r (0.98). CHIRPS and TAMSAT underestimated the observed rainfall while ARC2 indicated an overestimation. Additionally, CHIRPS and TAMSAT indicated the lowest RMSE of 14.16 mm and 14.14 mm whereas ARC2 presented a higher RMSE of 26.22 mm (Table 2).

In the Guinea Zone (Figure 3f), all the estimates were able to capture the annual cycles and a slight break of rain in August, which is sometimes called the little dry season [45,46]. The entire set of estimates show encouraging agreements with observed rainfall by indicating high r (>0.93), with TAMSAT producing the highest r (0.97). CHIRPS exhibited an underestimation while ARC2 and TAMSAT showed an overestimation (Table 2). The lowest RMSE value of 8.86 mm was obtained by TAMSAT, while ARC2 and CHIRPS indicated RMSE values of 42.53 mm and 9.21 mm, respectively (Table 2).

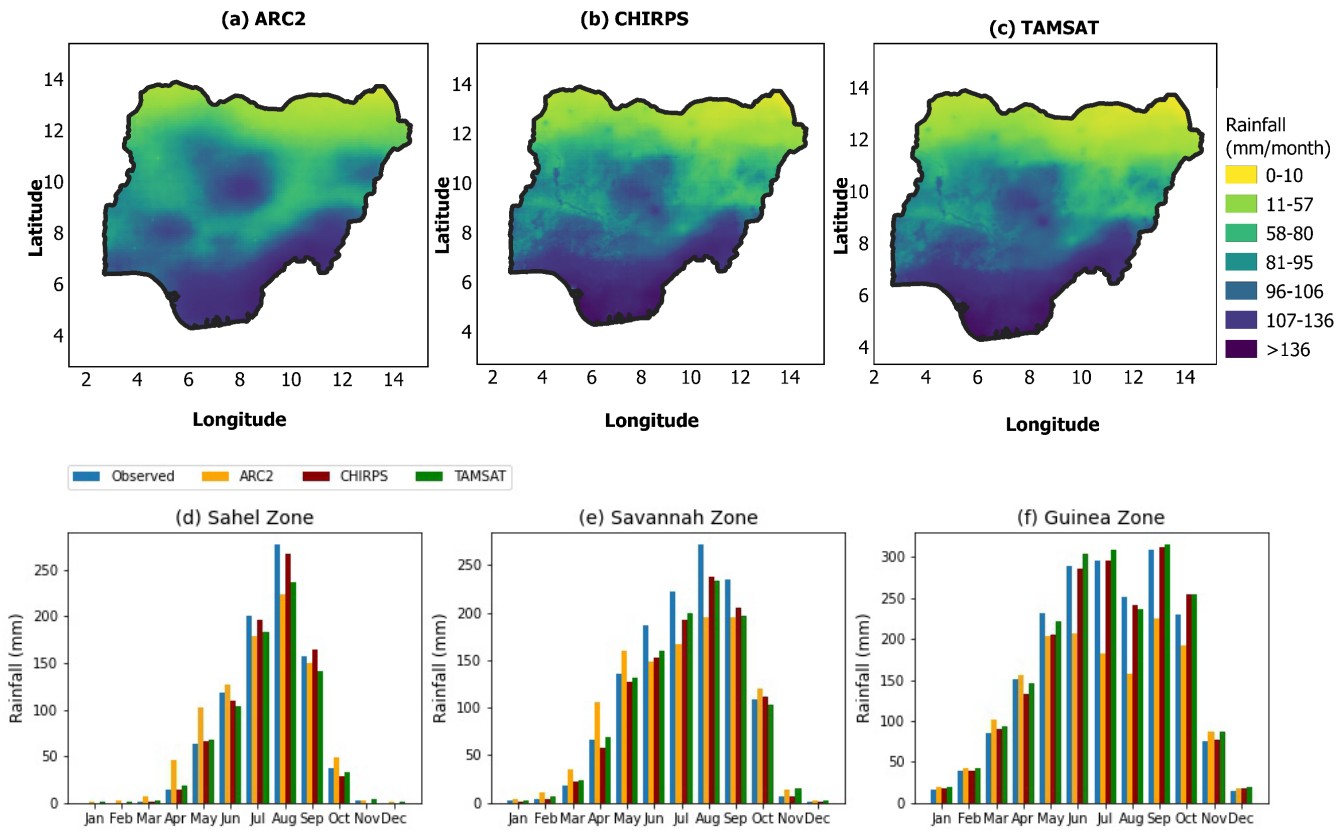

**Figure 3.** (**a**–**c**) Spatial distribution of mean monthly rainfall (mm/month) from 2002 to 2022, and corresponding annual cycles in (**d**) Sahel Zone, (**e**) Savannah Zone and (**f**) Guinea Zone for observed rainfall and SREs estimates (ARC2, CHIRPS and TAMSAT).

**Table 2.** Summary of the statistical indicators (correlation coefficient (r), bias, mean error (ME) and root mean square error (RMSE)) for mean monthly rainfall products.

|  | Sahel | | | Savannah | | | Guinea | | |
|---|---|---|---|---|---|---|---|---|---|
|  | ARC2 | CHIRPS | TAMSAT | ARC2 | CHIRPS | TAMSAT | ARC2 | CHIRPS | TAMSAT |
| r | 0.98 | 0.99 | 0.99 | 0.96 | 0.98 | 0.97 | 0.94 | 0.96 | 0.97 |
| Bias | 1.02 | 0.97 | 0.90 | 0.92 | 0.90 | 0.91 | 0.80 | 0.99 | 1.03 |
| ME | 0.75 | −1.17 | −4.0 | 4.86 | −6.52 | −5.47 | 18.82 | −0.85 | 3.08 |
| RMSE | 16.93 | 4.02 | 10.91 | 26.22 | 14.16 | 14.14 | 42.53 | 9.21 | 8.86 |

### *3.2. Performance Evaluation of SREs Detection Capability*

To understand the detection capabilities of the SREs (ARC2, CHIRPS and TAMSAT) with respect to observed rainfall, we computed the POD, FAR and CSI using average daily rainfall data across the stations of each agro-climatic zone (Sahel, Savannah and Guinea), covering the period 2002–2022. Quantiles 90 (Q90) were selected in order to evaluate the detective capability of these products at a high threshold.

Figure 4 shows the computed values of monthly bias (MB) for all of the data and the Q90 of each zone. Figure 4a shows the Sahel MB values when all the data are included in the analysis. All the products indicated an overestimation (MB > 1) in March, April, May and October (except for CHIRPS in October) while the bias values between June and September were around 1 (Figure 4a). The CHIRPS scored a close to perfect bias value from June to September and ARC2 scored the same in September. On the other hand, the monthly quantiles bias value (MQB) increased by about two times during March, April,

May and October (compare Figure 4a and Figure 4b). Between June and September, it is noted that the MQB values of all the products do not significantly change, but in November the bias values of ARC2 and TAMSAT exhibit overestimation.

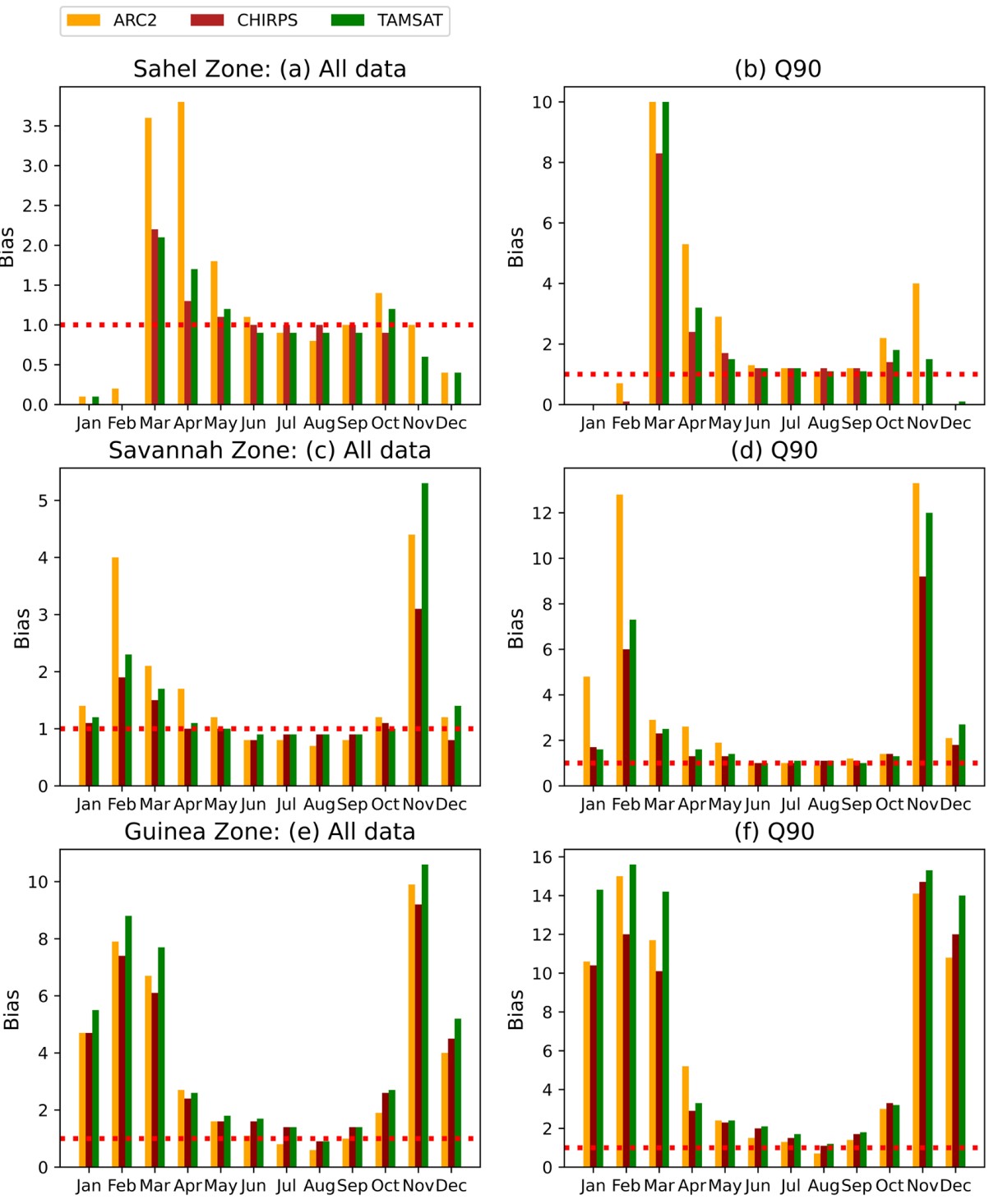

**Figure 4.** Monthly bias of three agro-climatic zones. Sahel Zone; (**a**) all data and (**b**) Q90, Savannah Zone; (**c**) all data and (**d**) Q90, Guinea Zone; (**e**) all data and (**f**) Q90. The red dotted line represents the perfect bias values (Bias = 1).

Figure 4c shows the MB values of all the estimates over the Savannah Zone when the all of data are included in the analysis was performed for the Sahel zone. The products exhibited overestimations (MB > 1) mostly during January, February, March, April, November and

December (except for CHIRPS in January, April and December) (Figure 4c). This is comparable to the bias values which are around 1 for the rest of the year. CHIRPS showed a better performance in most of the months, followed by TAMSAT. During January, February and November the MQB values increased to almost double (compare Figure 4c and Figure 4d). As shown in Figure 4d, it was observed that, for high thresholds, the MQB values of all of the estiamtes between May and October were around 1 (except for ARC2 in May).

Moreover, in the Guinea Zone, Figure 4e shows that all the products exhibited an overestimation between January and April and October and December. Similarly, the products showed less bias during the rest of the months. It is noted that all the SREs showed a higher bias in this zone than in both Sahel and Savannah Zones. As noted in the Sahel and Savannah Zones, the MQB values of all the products increased between January to April and October to December, whereas the products did not significantly change from May to September.

The monthly mean quantiles errors (MQEs) in the three agro-climatic zones are demonstrated in Figure 5a–c. Figure 5a shows the Sahel Zone MQEs for ARC2, CHIRPS and TAMSAT with respect to the Q90 of observed rainfall. As shown, all the products tended towards high significant overestimation, particularly between April and October, with TAMSAT indicating lower mean error followed by CHIRPS and then ARC2 (Figure 5a).

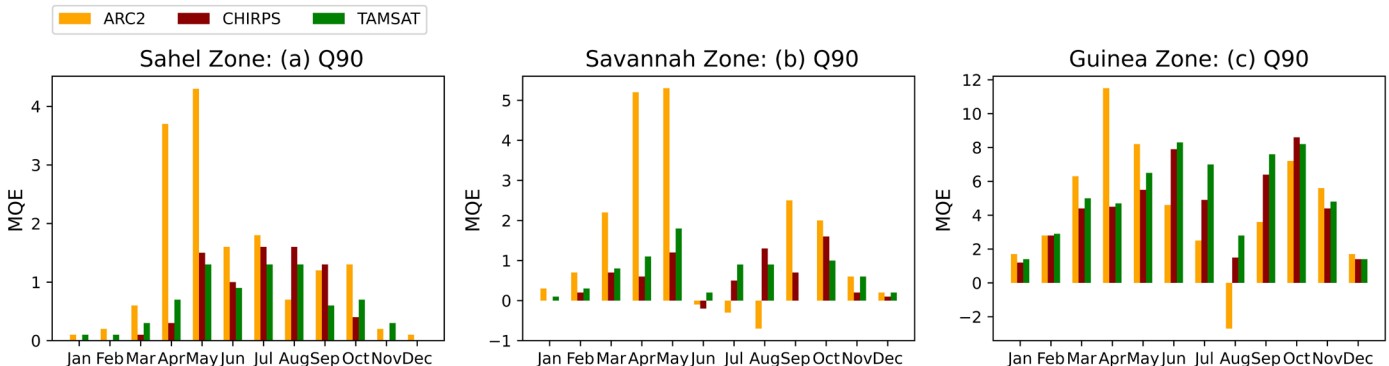

**Figure 5.** Monthly mean quantiles error (MQE (mm/month)) with respect to Q90 of observed rainfall for (**a**) Sahel Zone, (**b**) Savannah Zone and (**c**) Guinea Zone.

Similarly, the MQE over the Savannah Zone is presented in Figure 5b. All the products indicated considerable overestimation between March and October (except for ARC2 and CHIRPS which underestimated the observed rainfall in June, July and August). It is noted, in this zone, that all three estimates showed more overestimation than underestimation. Additionally, all the estimates exhibit a lower mean error in this zone between June and August than in the Sahel Zone (Figure 5b). Moreover, the mean error values obtained over the Guinea Zone are larger than the results obtained in both the Sahel and Savannah Zones (compare Figure 5a–c). The entire product overestimated observed rainfall in the Guinea Zone (except for ARC2 which indicates underestimation in August). Generally, CHIRPS and TAMSAT had lower mean error values than ARC2 in all of the agro-climatic zones (Figure 5).

The probability of detection (POD) values for all of the data and Q90 are presented in Figure 6a–f for all three agro-climatic zones. In the Sahel Zone (Figure 6a), the best POD values were obtained by the three products from May to September (MJJAS) which covers a period of major rainfall over Sahel. Conversely, the SREs exhibit low POD values in November, December, January, February and March (NDJFM). Additionally, all the SREs indicated an increase at the Q90 threshold (Figure 6b). CHIRPS showed a better performance by scoring a perfect value (1) for the QPOD between April and October, followed by ARC2 from April to September and then TAMSAT in May to September.

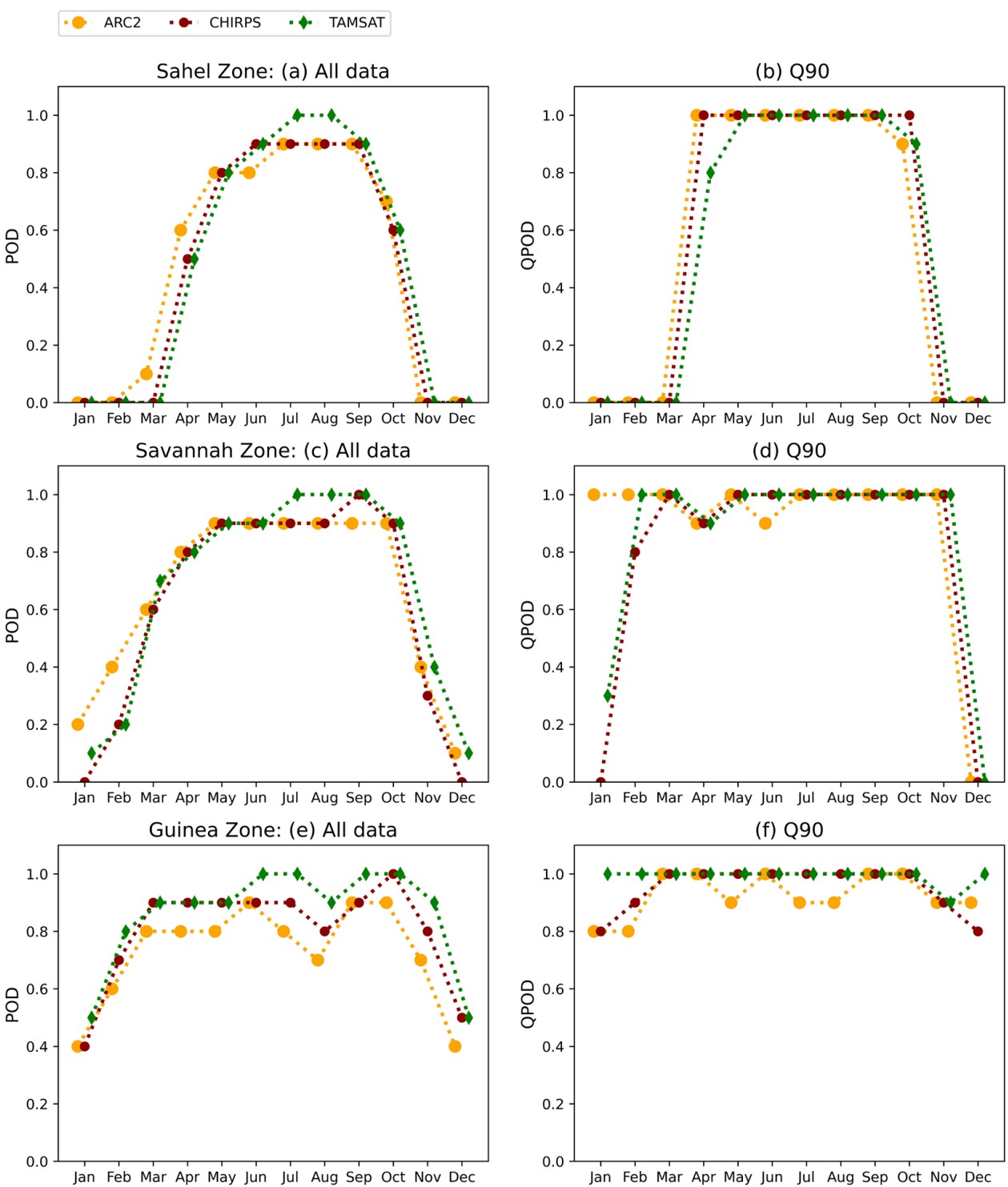

**Figure 6.** Probability of detection (POD) of three agro-climatic zones. Sahel Zone: (**a**) all data and (**b**) Q90; Savannah Zone: (**c**) all data and (**d**) Q90; Guinea Zone: (**e**) all data and (**f**) Q90.

All the estimates showed lower FAR values (Figure 7a) from June to September, and high FAR values in March, April, May, October and November. The QFAR values of all of the SREs showed an increase (Figure 7b). As shown, all the estimates exhibited lower QFAR values from June to September. CHIRPS and TAMSAT exhibited high QFAR values in March, April, October and November (except for CHIRPS in November) whereas ARC2 showed high QFAR values between January to April and October to November. Additionally, for high thresholds (Q90), the QFAR of all the products is higher over November,

March and April than the results shown in Figure 7a (all data). This implied that, with respect to a high threshold (Q90), the FAR is higher during the dry season. Overall, the ARC2 and TAMSAT estimates are subject to higher FAR values than the CHIRPS estimates. Moreover, Figure 8a,b support the findings of both Figure 6a,b and Figure 7a,b by indicating high CSI and QCSI values between June and September and weak CSI and QCSI values during NDJFM.

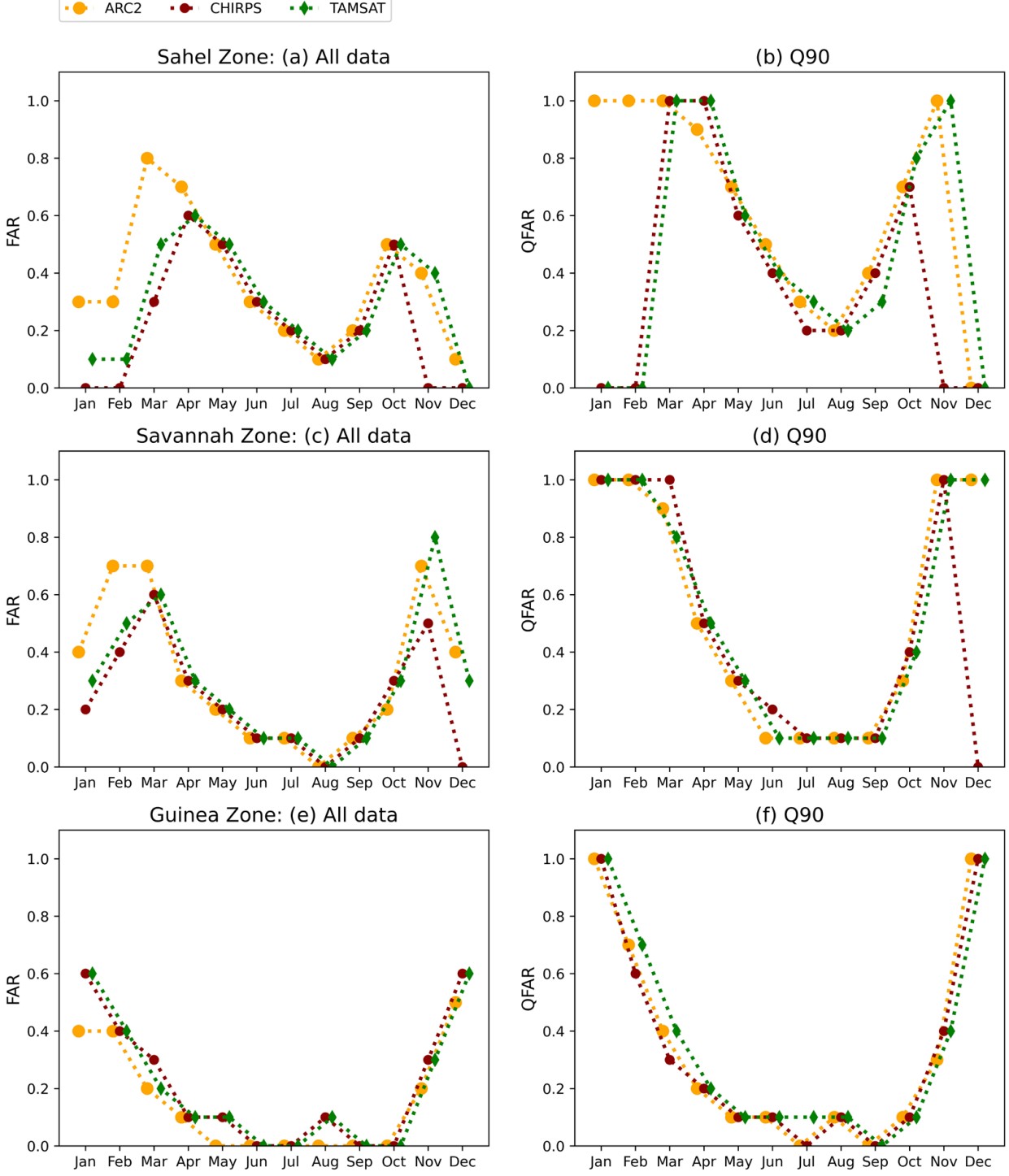

**Figure 7.** False alarm ratio (FAR) of three agro-climatic zones. Sahel Zone: (**a**) all data and (**b**) Q90; Savannah Zone: (**c**) all data and (**d**) Q90; Guinea Zone: (**e**) all data and (**f**) Q90.

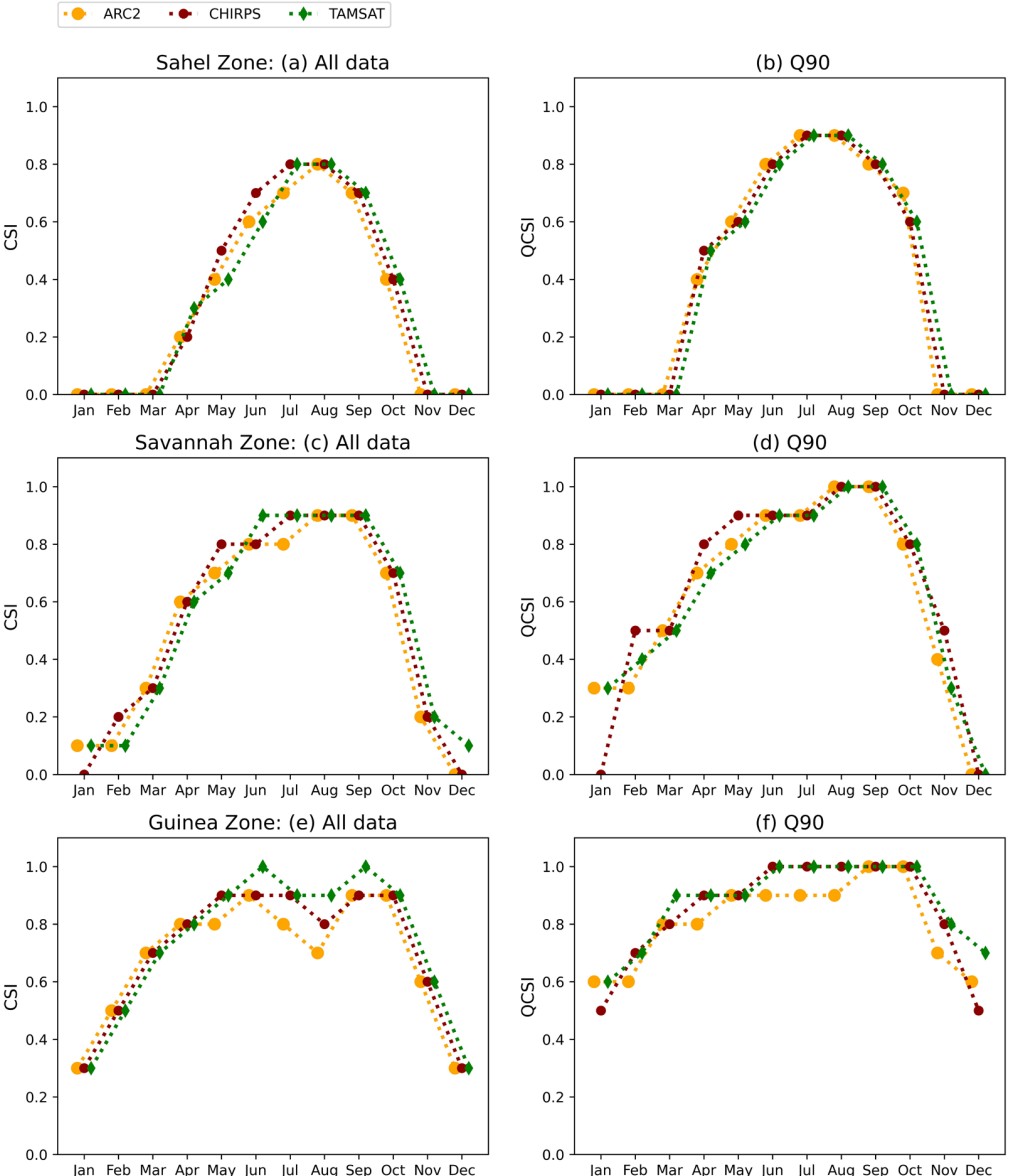

**Figure 8.** Critical success index (CSI) of three agro-climatic zones. Sahel Zone: (**a**) all data and (**b**) Q90; Savannah Zone: (**c**) all data and (**d**) Q90; Guinea Zone: (**e**) all data and (**f**) Q90.

Figure 6c,d demonstrates the POD (all data) and QPOD over the Savannah Zone. Similar to the results found in the Sahel Zone, the SREs show a slight increase in the QPOD values at the Q90 threshold (compare Figure 6c,d). Figure 6c indicates that the three products exhibit high POD values between May and October (MJJASO), regardless of the choice of threshold. A lower performance is shown by all of the estimates between November and March (NDJFM) (Figure 6c). Similarly, all the products exhibit high QPOD values between January and November (except for CHIRPS and TAMSAT in January) (Figure 6d). However, as shown in Figure 7c, all products exhibit lower FAR values between April and October, and high FAR values during November to March (NDJFM). For the high threshold (Q90), the QFAR was higher during NDJFM except for CHIRPS which indicated a perfect QFAR value in December (Figure 7d). This shows that FAR is higher during the dry season than the finding given in Figure 7c (all data). Furthermore, Figure 8c,d shows correctly detected months where all the estimates indicated high CSI and QCSI values in MJJASO and low CSI and QCSI values during NDJFM.

Figure 6e,f presents the POD and QPOD values over the Guinea Zone. CHIRPS and TAMSAT exhibited high POD values between March and November, followed by ARC2

(Figure 6e). As shown, all the estimates showed lower POD values during December, January and February. CHIRPS and TAMSAT produced high QPOD values between March and November, followed by ARC2 (Figure 6f). It is worth mentioning that CHIRPS and TAMSAT outperformed ARC2 in exhibiting higher POD and CSI values (Figure 6e,f and Figure 8e,f). Moreover, all the SREs exhibited lower FAR values (Figure 7e) between March and November and higher values during December, January and February. Similarly, for the high thresholds (Q90), the QFAR values (Figure 7f) were higher during December, January and February than the finding in Figure 7e (all data). This means that, with respect to the high thresholds, the FAR is higher during the period of the dry season which is similar to the results found in the Sahel and Savannah Zones. Additionally, Figure 8e,f indicate higher CSI values between March and November whereas the estimates showed low CSI values in December, January and February. CHIRPS and TAMSAT produced higher CSI values than ARC2.

Seasonal Variability of Satellite Rainfall Algorithm Skills

In this section, the seasonal variability of the satellite-based rainfall algorithms' skills is presented. Figure 9a–l shows a summary of the skills of all three SREs with respect to different metrics during peak rainfall periods and dry periods with low or no rainfall over the Sahel Zone (Figure 9a–d), Savannah Zone (Figure 9e–h) and Guinea Zone (Figure 9i–l). The reader will notice that 1-FAR is plotted instead of FAR. This is applied to make the best score for all indicators 1. Figure 9 was created by calculating the aforementioned statistics based on the peak rainfall periods and dry periods with low or no rainfall within the dataset used (2002–2022). As shown, Figure 9a–l clearly presents the algorithm's performance during the two climate conditions of the Sahel, Savannah and Guinea Zones.

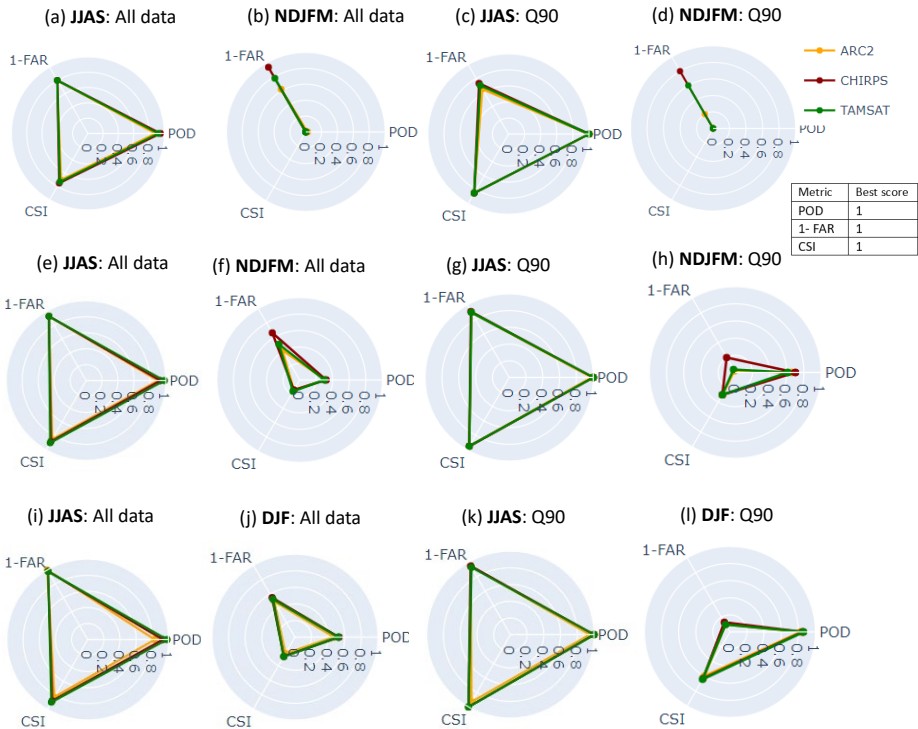

**Figure 9.** Variability of rainfall algorithms' skills in detecting rainfall, during peak rainfall period (JJAS) and dry seasons (NDJFM and DJF) of all data and Q90 over (**a–d**) Sahel Zone, (**e–h**) Savannah Zone and (**i–l**) Guinea Zone.

In the Sahel Zone, as depicted in Figure 9a (all data), the POD values for ARC2, CHIRPS and TAMSAT during JJAS are 0.88, 0.95 and 0.92, respectively, whereas the CSI values are 0.70, 0.75 and 0.73, respectively, indicating that CHIRPS has better skills to correctly detect more rainfall events. Similarly, at the Q90 threshold, the POD and CSI

values do not significantly change (Figure 9c). In addition, the 1-FAR values of ARC2, CHIRPS and TAMSAT are 0.80, 0.81 and 0.80, indicating that CHIRPS has the lowest FAR, followed by ARC2 and TAMSAT (Figure 9a). As shown in Figure 9c, the 1-FAR values of all products do not change greatly at the Q90 threshold. Moreover, during NDJFM (dry period) the POD and CSI values of all SREs were almost 0 for both the all data and Q90 thresholds (Figure 9b,d). This implies that all of the algorithms exhibited poor detection skills in the dry period. The 1-FAR values of all three SREs decreased at the Q90 threshold (compare Figure 9b and Figure 9d), which shows that all of the estimates missed more rainfall events at a higher threshold, with CHIRPS indicating the lowest FAR, followed by TAMSAT and then ARC2.

Over the Savannah Zone (Figure 9e), during JJAS, ARC2, CHIRPS and TAMSAT exhibited respective POD values of 0.90, 0.93 and 0.96, with corresponding CSI values of 0.85, 0.89 and 0.90. This suggests that CHIRPS and TAMSAT are more effective in accurately identifying rainfall events across all of the data. Similarly, there is little change in the POD and CSI values at the Q90 threshold (Figure 9g). Additionally, ARC2, CHIRPS and TAMSAT demonstrate 1-FAR values of 0.92, 0.93 and 0.93, respectively, implying that CHIRPS and TAMSAT have the lowest FAR, followed by ARC2 (Figure 9e). Figure 9g illustrates minimal variation in 1-FAR values across all of the estimates at the Q90 threshold. Furthermore, in the dry period ARC2, CHIRPS and TAMSAT scored POD values of 0.31, 0.32 and 0.30, respectively, and the CSI values were 0.14, 0.14 and 0.16 (Figure 9f). ARC2 and CHIRPS exhibited the highest POD values whereas TAMSAT had the highest CSI value. At the Q90 threshold, the POD and CSI values of all the estimates increased (Figure 9f,h). On the other hand, the 1-FAR values of all the estimates were reduced at the Q90 threshold (see Figure 9f,h), indicating that all of the estimates created a high number of false alarms during the dry period. CHIRPS and TAMSAT showed the lowest FAR values followed by ARC2.

Moreover, in the Guinea Zone (Figure 9i), the POD values for ARC2, CHIRPS and TAMSAT during JJAS are 0.83, 0.95 and 0.97, respectively. Meanwhile, their respective CSI values are 0.83, 0.88 and 0.90. This indicates that CHIRPS and TAMSAT are more proficient in accurately detecting rainfall events based on all of the data. Similarly, minimal changes are observed in both the POD and CSI values at the Q90 threshold, as depicted in Figure 9k. ARC2, CHIRPS and TAMSAT exhibit 1-FAR values of 0.99, 0.98 and 0.98, respectively (Figure 9i), indicating that all of the estimates had the lowest FAR values during JJAS. Figure 9k shows that the 1-FAR values of all products do not significantly change at the Q90 threshold. In addition, during the dry period (DJF), ARC2, CHIRPS and TAMSAT achieved POD values of 0.47, 0.53 and 0.52, respectively, while their CSI values were 0.23, 0.27 and 0.27, as shown in Figure 9j (all data). CHIRPS and TAMSAT exhibited the highest POD and CSI values followed by ARC2. Similarly, when considering the Q90 threshold, both the POD and CSI values of all products increased (compare Figure 9j and Figure 9l). Conversely, the 1-FAR values of all products decreased at the Q90 threshold (Figure 9j,l), indicating a high ratio of falsely detected rainfall events during the dry period for all of the estimates. CHIRPS exhibited the lowest FAR, followed by ARC2 and TAMSAT. Generally, the seasonal variability of the rainfall algorithms' skills revealed that the SREs generally performed better during the wet season (JJAS) compared to the dry season (NDJFM), with CHIRPS exhibiting the lowest false alarms values during NDJFM, followed by TAMSAT. This suggests that the low performance of the SREs during dry season could be attributed to the rainfall retrieval algorithms, intensity of rainfall occurrence and spatial-temporal resolution, which potentially lead to low accuracy in the rainfall retrieval carried out by the algorithms resulting from the intense stratiform cloud coverage [1,42,47–50].

## 4. Discussion

In this study, we utilized a point-to-pixel evaluation approach to assess the accuracy of SREs against gauge observations. Among the three products examined, CHIRPS exhibited the best performance, followed closely by TAMSAT. These findings align with the previous studies [51–56] conducted over West African regions and East Africa [38,57]. Similarly,

CHIRPS and TAMSAT outperformed ARC2 in terms of capturing the southern–northern latitude oscillations in the ITCZ, which led to convective processes in all of the agro-climatic zones, including in the representations of both unimodal and bimodal annual rainfall patterns of Nigeria noted by [2]. In addition, CHIRPS performed relatively better in detecting rainfall occurrence and had less bias during the dry season, indicating its robustness in differentiating between rainy and dry conditions. These results align with the conclusions drawn by [2,51] regarding CHIRPS. ARC2 showed a poor performance in another recent study [55] conducted over the West African region. The poor skills of ARC2 could be related to the rainfall retrieval algorithm, the merging process and source of the data used during its generations [50]. Moreover, with respect to a high threshold (Q90), the false alarm ratio of all of the estimates was higher during the dry season than wet season, with CHIRPS experiencing the least false alarms during dry season, followed by TAMSAT. This is similar to the results of previous studies [51] conducted over West Africa and China [23,43], which discovered that CHIRPS missed less rainfall during the dry season. However, further research is still needed to scrutinize whether CHIRPS performs better in detecting rainfall occurrence in the dry season.

All the estimates indicated a larger mean error in the Guinea Zone than in both the Sahel and Savannah Zones. This variation may be attributed to the diverse land use and land cover in the region, as well as the presence of the Atlantic Ocean in the southernmost part of the country, which affects rainfall patterns [33]. A similar study [19] revealed that the Guinea Zone naturally experiences frequent cloud cover, which remains consistent regardless of rainfall. This persistent cloud cover could pose difficulties for satellite retrieval. Conversely, cloud presence in the Sahel and Savannah Zones is primarily linked to rainy episodes, reducing the likelihood of discrepancies between clouds associated with rainfall and those without. This variation may contribute to the stronger correlation observed over the Sahel Zone.

In this research, despite carrying out comprehensive quality control measures, the mismatch between the point (gauge stations) and pixel (SREs estimate) measurements could underestimate the actual performance of SREs [25,29]. Nevertheless, the study by Zhang et al. (2018) [58] proposed that assessments conducted using either point-to-pixel or pixel-to-pixel methodologies yielded comparable statistical outcomes, indicating that the reciprocal evaluation and ranking of SREs may be disregarded. In addition, future work could focus on the evaluation of satellite-retrieved extreme rainfall rates over Africa. A thorough understanding of the rainfall rates obtained from satellites is essential for enhancing early warning systems, forecasting and modeling, particularly in regions where gauge stations are scarce. This topic is interesting as it can also contribute to mitigating the significant impacts posed by extreme rainfall events on agriculture, environment and the economy [11].

## 5. Conclusions

In this study, three satellite products (ARC2, CHIRPS and TAMSAT) were evaluated with respect to rain gauge observation for the period 2002–2022. The results have important implications for weather monitoring and forecasting in the region. These are summarized below.

All of the SREs showed the ability to capture the spatial patterns of mean annual rainfall, with CHIRPS showing a high spatial resolution, r (0.92), bias (0.97) and lower RMSE (61 mm/annum), followed by TAMSAT. Similarly, the all of the products show a strong agreement (r > 0.94) with the observed data at mean monthly time steps. CHIRPS and TAMSAT showed the lowest RMSE values of 4.02 mm/month and 8.86 mm/month.

Moreover, in both the Sahel and Savannah Zones, all of the SREs indicated less bias during the major rainfall season (MJJASO) and mostly overestimation during the dry season (NDJFM), while in the Guinea Zone, the products exhibited less bias between May and September and more overestimation from January to April and October to December. In addition, the entire products exhibit larger mean error over the Guinea Zones than in

Sahel and Savannah Zones. Overall, CHIRPS and TAMSAT performed best, with less bias, followed by ARC2.

Additionally, all three SREs were skillful in detecting rainfall occurrence during the wet season in the Sahel and Savannah Zones, with high false alarm values largely occurring during the dry season. Similarly, the detection capability of all the products was also more encouraging in the wet season than in the dry season over the Guinea Zone. Generally, the products showed high detection capabilities and a lower false alarm ratio during the wet season than the dry season. CHIRPS and TAMSAT appear to have better detective capabilities during the dry season.

The findings of this study offer valuable insights into selecting appropriate satellite rainfall estimates (SREs) for regional applications, particularly in Nigeria. Moreover, our research underscores the importance of addressing the identified limitations of existing SREs and exploring innovative approaches to enhance their performance. By incorporating new methodologies and data sources, there is potential to mitigate the challenges associated with SREs and improve their utility for various applications, such as hydrological modeling and agricultural planning. In addition, the findings of this study could also help algorithm developers upgrade the performance of SREs' retrieval algorithms.

**Author Contributions:** Conceptualization, A.D.D., G.Z., E.T., R.C. and F.P.; methodology, A.D.D.; software, A.D.D.; formal analysis, A.D.D.; investigation, A.D.D.; data curation, A.D.D.; writing—original draft preparation, A.D.D.; writing—review and editing, A.D.D., G.Z., E.T., R.C., F.P., B.A.A. and A.O.; supervision, G.Z., E.T. and R.C.; funding acquisition, G.Z. All authors have read and agreed to the published version of the manuscript.

**Funding:** This research is supported by the National Key Research and Development Program of China (grant no. 2022YFF0801704).

**Data Availability Statement:** African Rainfall Climatology version 2 (ARC2) was retrieved from https://www.ftp.cpc.ncep.noaa.gov (accessed on 10 July 2023); Climate Hazards Center InfraRed Precipitation with Station data (CHIRPS) is available at the website of University of California (https://www.chc.ucsb.edu/data/chirps, accessed on 10 July 2023), and the Tropical Application of Meteorology using Satellite data and ground-based observation (TAMSAT) can be obtained from https://www.tamsat.org.uk/data/ (accessed on 20 July 2023).

**Acknowledgments:** The authors thank the High-Performance Computing Center of Nanjing University of Information Science & Technology provides support for the data processing and visualization of this study. We would also particularly like to thank the Nigerian Meteorological Agency (NiMet) for providing rain gauge observation data.

**Conflicts of Interest:** The authors declare no conflicts of interest.

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
