# Peer review of "Evaluation of Satellite-Based Rainfall Estimates against Rain Gauge Observations across Agro-Climatic Zones of Nigeria, West Africa"

_remotesensing, doi:10.3390/rs16101755_

Round 1

Reviewer 1 Report

Comments and Suggestions for Authors

The manuscript „Evaluation of satellite-based rainfall estimates against rain 2 gauge observations across agro-climatic zones of Nigeria, West 3 Africa.“ compares three methods estimating precipitation using satellite data for agro-climatic zones of Nigeria. The topic is interesting not only from practical point of view.

My general comments are:

If you use values of RMSE, please add also mean values in order to evaluate the size of RMSE. I did not find values of Q90.

Comments

L17 – Unfortunately, it is known that satellite based precipitation estimates are not very accurate and reliable therefore the formulation should be modified.

L40 – “last recent” is not good wording

L40 – “These” – this sentence should be reformulated. I think it is not correct.

L64 – Really “prediction”?

L80 – What do you mean by “detection capability”.

L99 – Do you mean 235 K?

L138 – “than Sahel Zone”, another wording is needed.

L139 – What do you mean by “abrupt”. Please explain.

L172 – Please explain differences and meaning between n and N and variables with stripes.

L179 – Please express the sentence in more details. What do you mean by product?

P197 – The mentioned variables are not in equations.

L204 – observations

L210 – Is it lower or lowest?

L215 – What is “quietly”?

L223 – “indicated” – here and also in other places in paper I recommend using past tense.

L224 – Please reformulate the sentence. It is not clear relation to convective processes, please explain.

L364 – What is “detecting rainfall”?

Author Response

Please see the attached reviewer's comments.

Reviewer 2 Report

Comments and Suggestions for Authors

This is an interesting and, I think, useful study. Overall, the narrative is generally well signposted, so the various steps in analysis are clear. It is good to see the meaning of the statistical parameters explained. This helps the reader more easily interpret the value of the results.

However, there are quite a few issues with the presentation and, hence, the interpretation of the findings, which need to be addressed. The text also needs a thorough review and edit for editorial and grammatical inconsistencies and errors. If a review of the MS is permitted I suggest that it is checked by a native English speaker prior to resubmission.

Lines 17, 38, 45, 62, 64, 82, 142, 206, 221, 254, 260, 264, 274, 277, 288, 297, 310, 319, 322, 327, 352, 357, 358, 368, 377, 382, 403, 407, 458: delete the comma before ‘and’. In this context it is grammatically incorrect/redundant.

Line 21 (and elsewhere): check acronym capitalisation for TAMSAT. The first three letters of 'satellite' should also be capitalised

Line 24: you have said ‘include’. What other metrics were used?

Lines 29 : you need a capital Z for ‘zones’.

Line 29: insert ‘the’ before ‘dry’.

Lines 27-29: I assume this overestimation is due to issues with stratiform vs convective rainfall? This was my first impression reading the Abstract, although the possibility does not appear to be (clearly) drawn out in the narrative. Either way, the reason for the mismatches between i served and estimated rainfall should be flagged in the Abstract.

Line 40: ‘last recent decades’ is a cumbersome expression. I suggest just giving the dates.

Line 51: change ‘conclude’ to ‘concluded’.

Line 53: threshold of what?

Line 56: replace ‘compares’ with ‘agrees’.

Line 61: change ‘assess’ to ‘assessed’.

Line 62: delete ‘of’.

Line 63: change ‘discover’ to ‘discovered’.

Line 63: change ‘as’ to ‘gave’.

Line 66: change ‘show’ to ‘showed’.

Line 72: insert comma after ‘Nigeria’.

Line 74: move ‘effectively’ to after ‘services’.

Line 76: delete ‘i.e.’.

Line 79: ‘aims’ implies uncertainty. Delete ‘aims to’ and change ‘identify’ to ‘identifies’.

Line 98: delete ‘of’.

Lines 102, 109, 120: font change for the website addresses.

Lines 107-108: this is (worryingly) imprecise and incorrect. ‘Pentadal’ means 5-day, not 5-daily, there is an important difference! ‘Decadal’ means 10-days! What was ‘approximate’ about the sampling. If there is variation around 10 days this makes any calculations based on the data unreliable and non-comparable. Overall, this section of text needs careful review and editing.

Lines 124, 154: just give the number as numerals, ‘42’.

Lines 125-126: how many are ‘some’ stations? The reader needs to know the proportion of stations covering different years within the survey period, and how these variations were addressed in the analysis.

Line 129: you need to either give the guidelines (preferred) or give a reference so that the reader can find them.

Line 131: the right hand edge of the figure has been cut off. In the legend, ‘meters’ should be ‘metres’.

Line 132: insert ‘the’ before ‘Sahel’, ‘Savannah’ and ‘Guinea’ and ‘Zone’ after each of these names.

Lines 138-139, 141, insert ‘the’ before the names of the zones.

Lines 138, 140, 143, 221, 232, 237, 271, 272, 280, 284, 295, 301, 302, 307, 336, 373, 388, 389, 411, 412, 438, 446: capital ‘Z’ required for ‘Zone(s)’. These are the instances I have picked up, the MS should be checked carefully for any I have missed.

Lines 140-141: replace full stop with a comma and change ‘While’ to ‘while’.

Line 141: delete comma after ‘process’.

Line 142: insert ‘The’ before ‘Rainfall’ and change the latter to ‘rainfall’.

Line 143: insert ‘the’ before ‘Guinea’.

Lines 148-150: the text from the comma after ‘Nigeria’ can be deleted (the information appears in the legend of Figure 1) and the remaining text from line 147 onwards moved to the head of Section 2.3.

Line 156: what does ‘slightly underestimated ‘ mean? This needs quantification (and discussion of any implications).

Lines 159-160: sentence needs rewording to avoid starting it with ‘r’.

Line 166: add ‘respectively’ after ‘0’ and give a reference.

Line 172, 197: change ‘Where’ to ‘where’.

Line 176-177: check capitalisation. 

Lines 191-199, 248-249: font changes in table and equations need correcting for consistency with rest of MS.

Line 192: give units of rainfall rate. What is the threshold?

Line 198: insert ‘, respectively, ‘ after ‘detections’.

Line 210: you have introduced rainfall rates as per annum here. Previously you had defined them based on daily rates, and elsewhere in the text you refer to monthly rates. I can’t see any clear narrative explaining how you move from daily to monthly to yearly measurements, or the implications if this. If it is in the text, I have missed it. Either way, it needs a much clearer explanation (and I am not sure of the validity of this approach).

Line 210, 230, 231, 236, 241, 242, 437: space needed between number and units.

Lines 210-211: replace full stop with a comma, and change ‘Followed’ to ‘followed’.

Lines 215-216: should be ‘Intertropical Convergence Zone’.

Line 217: the legend for Figure 2 is meaningless! The same values of rainfall rates could appear in more than one category (eg 791, 1040, 1244, 1389, 1563). This is worrying in the context of the analyses presented. How can we trust the results with this method of presenting the data? At the very least, this indicates sloppy checking of the analysis/presentation of the results, at worst it indicates a complete misunderstanding on the part of the author(s). Also, why were these intervals of rainfall chosen for display?

Line 225: insert ‘the’ before ‘ITCZ’.

Line 228: change ‘zones’ to ‘Zone’ and replace ‘the entire’ with ‘all’.

Line 235: you need to provide evidence if the overestimation. Refer to a figure or table.

Line 238: when was the slight break of rain? It is not clear from Figure 3.

Line 239: insert ‘set of’ after ‘entire’.

Line 244: there is a similar issue with the legend for Figure 3 as noted above for Figure 2.

Line 246: insert ‘Zone’ after the areas.

Line 248: the legend is poor. Legends need to stand alone: the acronyms and symbols all require definition.

Line 255: insert ‘the’ after ‘across’ and change ‘zones’ to ‘zone’: changed ‘covered’ to ‘covering’.

Line 261: replace ‘to’ with ‘and’.

Lines 268, 315, 330, 333: insert ‘Zone’ after each zone name in the figure headers.

Lines 269-270: what does the red dotted line indicate (I presume it is a value of MQB = 1) and what is its significance? This information needs to appear in the legend.

Line 277: change ‘compares’ to ‘compare’.

Line 283: insert ‘Zones’ after ‘ Savannah’.

Line 288: change ‘Sahel’s’ to ‘Sahel Zones’’.

Line 292: the vertical axis is labelled as MAE, the legend says MQE. Which is it?

Line 294: insert ‘Zone’ after each region name.

Line 298: delete ‘well’.

Line 299: we need a reference to a figure (or other information) to see the over/under estimation.

Line 300: insert ‘the ‘ before ‘Sahel’.

 Lines 301: insert ‘the’ before ‘Guinea’ and ‘Sahel’.

 Line 304: delete ‘appear to’.

 Line 306: change ‘were’ to ‘are’.

 Line 324: insert comma after ‘)’.

 Line 327: capital ‘F’ required for ‘figure’.

 Line 336: change text to ‘..found in the Sahel Zone, the..’

 Line 353: delete ‘The’.

 Line 361: insert ‘the’ before ‘Sahel’ and ‘Zones’ after ‘Savannah’.

 Line 368: insert ‘the’ before ‘Sahel’ and ‘Zone’ after each region name.

 Line 369: insert a full stop after second ‘FAR’ and change ‘this’ to ‘This’.

 Line 370: change ‘Figures 9 are’ to Figure 9 is’.

 Line 373: insert ‘the’ before ‘Sahel’.

 Line 374: insert ‘Zones’ after ‘Savannah’.

 Line 377: insert comma after ’JJAS’.

 Line 386: I don’t understand Figure 9. How are the results interpreted? The reader needs this information.

 Line 393: delete ‘the’.

 Line 403: insert ‘the’ before ‘merging’.

 Line 406: change ‘alarm’ to ‘alarms’.

 Line 409: insert ‘the’ before ‘dry’.

 Lines 412-415: vague. A much clearer and more detailed discussion of these influences on rainfall is required (in this context just sending the reader to references is inadequate). Also, within the Discussion, I suspect some of the poorer correlations between satellite and ground data will be due to different rainfall regimes (convective vs stratiform) at different times in the rainy seasons. This needs consideration when analysing the data and also discussing them.  

Line 419: cumbersome referencing style (which also indicates problems with numbering references).

Line 440: replace full stop with a comma and change ‘While’ to ‘while’.

Line 445: delete ‘the’.

Line 446: insert ‘the’ before ‘wet’ and delete ‘a’.

Comments on the Quality of English Language

See notes above.

Author Response

Please find the attached reviewer's comments

Round 2

Reviewer 2 Report

Comments and Suggestions for Authors

My thanks to the authors for addressing my comments so comprehensively. There are, however, a few further issues (mainly editorial) which require attention. It appears that my recommendation to have any revision checked by a native English speaker was not followed!

Line 29: insert ‘the’ after ‘over’

Lines 31-32: this seems counterintuitive. Do you mean convective or stratiform? Stratiform cloud is more likely in the dry season.

 Line 56: extreme value of what?

 Line 66: insert ‘that’ before CHIRPS.

 Line 67: delete ‘a’.

 Line 72: delete apostrophe in SRE’s.

 Line 83: ‘Among others’ is vague. You either need to give all the measures, or, delete this text.

 Line 154: delete ‘In this study’ and start sentence with ‘We’.

 Line 159: either give all the metrics or delete ‘such as’.

 Line 181: replace ‘this’ with ‘these’.

 Line 202: change ‘scale’ to ‘scales’.

 Line 205: delete 2nd ‘the’.

 Line 22: change ‘shows’ to ‘show’.

 Line 225: ‘leading to ‘ should be ‘bringing’.

 Line 237: capital ‘A’ required for ‘august’.

 Line 264: insert ‘by’ after ‘increases’.

 Lines 268/294: compared to the others, the clarity of these figures is poor (fuzzy).

 Line 326: delete comma after ‘March’.

 Line 346: replace ‘the entire’ with ‘all’.

 Line 369: the additional text helps, but I still do not understand Figure 9.

 Line 382: insert comma after ‘FAR’.

 Line 386: delete 2nd ‘the’ and ‘period.

 Line 397: delete ‘the’ and ‘period’.

 Line 398: replace ‘s’ with ‘are’.

 Line 479, 480, 481: insert ‘the’ before ‘Sahel’, ‘dry’ and ‘Guinea’, respectively.

Line 486: delete comma after ‘particularly’.

Comments on the Quality of English Language

Please see above

Author Response

Please find the attached reviewer comments.
